# A Male Subject with Congenital Adrenal Hyperplasia due to 21-Hydroxylase Deficiency Which Was Diagnosed at 31 Years Old due to Infertility

**DOI:** 10.3390/diagnostics13030505

**Published:** 2023-01-30

**Authors:** Hideaki Kaneto, Hayato Isobe, Junpei Sanada, Fuminori Tatsumi, Tomohiko Kimura, Masashi Shimoda, Shuhei Nakanishi, Kohei Kaku, Tomoatsu Mune

**Affiliations:** Department of Diabetes, Endocrinology and Metabolism, Kawasaki Medical School, 577 Matsushima, Kurashiki 701-0192, Japan

**Keywords:** congenital adrenal hyperplasia, 21-hydroxylase deficiency, adrenocorticotropic hormone, cortisol, 17α-hydroxyprogesterone, male infertility

## Abstract

Introduction: Congenital adrenal hyperplasia is caused by deficiencies in a number of enzymes involved in hormone biosynthesis in the adrenal glands or sexual glands. Adrenocorticotropic hormone (ACTH) secretion is enhanced by decreased cortisol production, leading to adrenal hyperplasia. The frequency of 21-hydroxylase deficiency (21-OHD) was the highest among congenital hyperplasias, and in 1989 it became one of the target diseases for newborn screening in Japan. Case presentation: A 31-year-old Japanese male visited our institution due to infertility. On admission, his height was 151.7 cm (average ± SD in the same age, sex and population: 172.1 ± 6.1 cm). It was noted that his height had not changed since he was ten years old, and that pubic hair was observed when he was 7 years old. He had azoospermia and his gonadotropin level was low. He had low levels of both luteinizing hormone (LH) and follicle-stimulating hormone (FSH) but high levels of free testosterone. He had a low cortisol level and high ACTH level. Abdominal computed tomography (CT) showed swelling of bilateral adrenal glands, although morphology was normal. Based on these findings, he was diagnosed with primary adrenal insufficiency and admitted to our institution. His height had not changed since he was ten years old. In addition, pubic hair was observed when he was 7 years old. His sexual desire was decreased, although he had no general malaise or fatigue. He did not have pigmentation of the skin, genital atrophy or defluxion of pubic hair, although his body hair was relatively thin. In endocrinology markers, ACTH level was high (172.2 pg/mL) (reference range: 7.2–63.3 pg/mL), although his cortisol level was 6.9 μg/dL (4.5–21.1 μg/dL). These data suggest that he suffered from primary adrenal insufficiency. LH and FSH levels were both low, but free testosterone and estradiol levels were high. These data excluded the possibility of central hypogonadism. Furthermore, the level of 17a-hydroxyprogesterone, a substrate of 21-hydroxylase, and the level of pregnanetriol, a metabolite of progesterone in urine, were both markedly high. Based on these findings, we ultimately diagnosed this patient with 21-hydroxylase deficiency. Conclusions: We experienced a case of congenital adrenal hyperplasia due to 21-hydroxylase deficiency which was diagnosed in a 31-year-old male with infertility. Therefore, the possibility of 21-hydroxylase deficiency should be borne in mind in subjects with infertility who were born before 1989 and who had not undergone newborn screening for this disease.

## 1. Introduction

Congenital adrenal hyperplasia is caused by deficiencies in a number of enzymes involved in hormone biosynthesis in the adrenal glands or sexual glands. Adrenocorticotropic hormone (ACTH) secretion is enhanced by decreased cortisol production, leading to adrenal hyperplasia. The frequency of 21-hydroxylase deficiency, which is one of the autosomal recessive disorders [1,2,3,4,5], is the highest among congenital adrenal hyperplasias, and it accounts for approximately 90% of cases. In 1989, 21-hydroxylase deficiency became one of the target diseases in newborn screening in Japan [6,7]. This disorder is classified into classic and non-classic types [8,9,10,11,12]. In patients with the classic type, severe conversion failure to cortisol and/or aldosterone after birth and severe adrenal deficiency are often observed. In contrast, in patients with the non-classic type, conversion failure is moderate. In general, in late-onset patients with the non-classic type, there are no symptoms at birth. The diagnosis of 21-hydroxylase deficiency is usually based on the following phenotypes: short stature, precocious puberty and adrenal incidentaloma. In addition, this disorder is sometimes diagnosed based on phenotypes such as masculinization in females and azoospermia or oligozoospermia in males.

We present a male patient with infertility who was diagnosed with primary adrenal insufficiency when he was 31 years old. He was born in 1988, and newborn screening for 21-hydroxylase deficiency started in Japan in 1989. Therefore, the presence of 21-hydroxylase deficiency, especially in subjects with infertility who were born before 1989, should be borne in mind.

## 2. Case Presentation

A 31-year-old Japanese male with infertility visited the Department of Obstetrics and Gynecology in our institution. Since he had azoospermia and his gonadotropin level was low, he was referred to the Department of Urology. Luteinizing hormone (LH) and follicle-stimulating hormone (FSH) levels were both low, but free testosterone levels were high, as follows: LH, <0.1 mIU/mL (reference range: 0.79–5.72 mIU/mL); FSH, 0.34 mIU/mL (2.00–8.30 mIU/mL); free testosterone, 25.5 pg/mL (6.5–17.7 pg/mL). He was referred to the Department of Diabetes, Endocrinology and Metabolism. His cortisol level was low (4.8 μg/dL) (4.5–21.1 μg/dL), and his ACTH level was high (93.9 pg/mL) (7.2–63.3 pg/mL). Based on these findings, he was diagnosed with primary adrenal insufficiency and admitted to our institution.

On admission, his height, body weight and body mass index were 151.7 cm, 70.2 kg and 30.7 kg/m^2^, respectively. Average ± SD in the same age, sex and population is 172.1 ± 6.1 cm). It was noted that his height had not changed since he was ten years old, and that pubic hair was observed when he was 7 years old. Blood pressure, heart rate and body temperature were 113/70 mmHg, 70 beats/min and 36.4 °C, respectively. He had no general malaise or fatigue, although his sexual desire was decreased. He did not have pigmentation of the skin, genital atrophy, or defluxion of pubic hair, although his body hair was relatively thin. There were no abnormalities in heart and lung sounds or in the abdomen. It was difficult to obtain information about family history for this patient because his parents were divorced soon after he was born, and he was raised by his grandparents. There were no endocrine disorders in his grandparents, but there was no contact address information for his parents; thus, we were unable to investigate whether his parents had any endocrine disorders. Table 1 shows the clinical parameters on admission for this subject. He had type 2 diabetes mellitus (HbA1c, 8.1%; fasting plasma glucose, 109 mg/dL) and several months before he had started taking 20 mg of tofogliflozin. Exacerbation of this patient’s type 2 diabetes mellitus was likely brought about by over-consumption of a high-carbohydrate diet. Liver dysfunction was observed as follows: AST, 54 U/L; ALT, 123 U/L; γ-GTP, 66 U/L. Renal function was within normal range, and HDL-C level was low (28 mg/dL).

In endocrinology markers, cortisol was 6.9 μg/dL (4.5–21.1 μg/dL) and ACTH was high (172.2 pg/mL) (7.2–63.3 pg/mL) considering the cortisol value. These data suggest that he suffered from primary adrenal insufficiency. Both LH and FSH were low (LH, <0.10 mIU/mL (0.79–5.72 mIU/mL); FSH, 0.22 mIU/mL (2.00–8.30 mIU/mL)), but free testosterone and estradiol were high (free testosterone, 29.7 pg/mL (6.5–17.7 pg/mL); estradiol, 63.1 pg/mL (14.6–48.8 pg/mL)). These data excluded the possibility of central hypogonadism. Also, we examined whether there was diurnal variation of ACTH and cortisol levels. In diurnal variation in healthy subjects, cortisol level is highest early in the morning (at 8:00) and cortisol level at midnight (at 23:00) is lower than 5 mg/dL. Considering from this criteria, diurnal variation of cortisol was preserved in this patient (8:00, 6.9 mg/dL; 14:00, 5.5 mg/dL; 22:00, 2.4 mg/dL; 23:00, 2.6 mg/dL). However, ACTH value was high at 8:00 (172 pg/mL) (7.2–63.3 pg/mL), and ACTH levels were relatively high at all points (14:00, 77.2 pg/mL; 22:00, 31.1 pg/mL; 23:00, 28.1 pg/mL) considering from the findings that diurnal variation of cortisol value was preserved. Next, we performed a rapid ACTH load test, after 30 min rest under fasting conditions in the morning, using 0.25 mg of tetracosactide acetate (timeline of assay: pre, 45 min, 60 min, 90 min, 120 min; measurement, CLEIA). In this test, cortisol response to ACTH was poor (cortisol: pre, 7.9 μg/dL; 45 min, 8.0 μg/dL; 60 min, 8.9 μg/dL; 90 min, 9.3 μg/dL; 120 min, 9.2 μg/dL). In corticotropin-releasing hormone (CRH), luteinizing hormone-releasing hormone (LHRH), growth hormone-releasing hormone (GRH), and thyrotropin-releasing hormone (TRH) loading test, there were no abnormalities in response for ACTH, LH, FSH, growth hormone (GH), thyroid-stimulating hormone (TSH) and prolactin (Table 2). However, cortisol response was very poor, and the cortisol level was not increased at all after the load test.

Computed tomography (CT) of the abdomen showed swelling of bilateral adrenal gland, although morphology of the adrenal glands was normal, suggesting the presence of adrenal hyperplasia (Figure 1). In addition, fatty liver, gallbladder stones and a right renal cyst were observed. There were no enlarged lymph nodes and no findings suggesting malignancy in the abdominal CT. Brain magnetic resonance imaging (MRI) showed no abnormality in the pituitary gland. 

Next, we examined the intermediate metabolites of steroid synthesis. As shown in Table 3, the level of 17α-hydroxyprogesterone, a substrate of 21-hydroxylase, was markedly high (42.7 ng/mL (reference value: 0.0–3.5 ng/mL)). In urinalysis, levels of pregnanediol and pregnanetriol, metabolites of progesterone in urine, were both high as follows: pregnanediol, 1.78 mg/day (0.16–0.79 mg/day); pregnanetriol, 36.12 mg/day (0.13–1.60 mg/day). In urine steroid profile analysis, pregnanetriol and 11-hydroxyandrrostenedione were both markedly high as follows: pregnanetriol, 10.1 mg/gCre (0.0–0.3 mg/gCre); 11-hydroxyandrrostenedione, 22.5 mg/gCre (0.0–0.35 mg/gCre). In addition, 17-ketogenic steroid (17-KGS), a metabolite of cortisol in urine, was low (0.67 mg/day (6.00–18.40 mg/day)). In contrast, many 17-ketosteroid (17KS) fractions were markedly increased as follows: androsterone, 7.18 mg/day (1.10–4.20 mg/day); 11-ketoandrosterone, 1.27 mg/day (0.00–0.12 mg/day); 11-ketoetiocholanolone, 3.00 mg/day (0.04–0.65 mg/day); 11-hydroxyandrosterone, 28.51 mg (0.40–2.30 mg/day). Based on these findings, we ultimately diagnosed this patient with 21-hydroxylase deficiency. 

After the diagnosis, we started 0.5 mg of long-acting steroid dexamethasone. Since ACTH level, especially in early morning, was relatively higher considering from cortisol level, we started long-acting steroid dexamethasone, but not hydrocortisone, in this patient. The ACTH level was substantially reduced to 3.9 pg/mL after starting the dexamethasone treatment. In addition, although this patient still suffers from infertility, the decreased sexual desire has been moderately mitigated. Although we recommended that this patient consulted a specialist in the Department of Genetic Medicine, he refused for economic reasons. He continued to visit our department for treatment of his type 2 diabetes mellitus. Although he was treated with 0.5 mg of dexamethasone, aggravation of glycemic control was not observed.

## 3. Discussion

The frequency of 21-hydroxylase deficiency is the highest among various congenital adrenal hyperplasias [1,2,3,4,5]. In Japan in 1989, 21-hydroxylase deficiency became one of the target diseases in newborn screening [6,7]. In this case report, we presented a 31-year-old Japanese male with infertility who was born before newborn screening for 21-hydroxylase deficiency began. This patient was ultimately diagnosed with primary adrenal insufficiency for the following reasons: low LH and FSH levels, high free testosterone level, low cortisol level and high ACTH level, and swelling of bilateral adrenal glands. The findings of low LH and FSH, and high free testosterone and estradiol, excluded the possibility of central hypogonadism. In urine steroid profile analysis, pregnanetriol and 11-hydroxyandrrostenedione were both high. Based on these findings, we ultimately diagnosed this patient with 21-hydroxylase deficiency.

The diagnosis of 21-hydroxylase deficiency in this case was based on phenotypes such as short stature, precocious puberty, azoospermia and adrenal incidentaloma. In general, in patients with the non-classic form, conversion failure to cortisol and/or aldosterone is not severe and there are no symptoms at birth. [8,9,10,11,12]. Indeed, this patient had no symptoms at birth, which is compatible with the non-classic form of this disease. 

Adult patients with 21-hydroxylase deficiency are often respond well to treatment with 5–10 mg hydrocortisone [13,14,15,16]. It is known, however, that the duration of action of hydrocortisone is relatively short; even when taken before sleep, its action is not long enough to suppress a surge in ACTH levels the following morning. The long-acting steroid dexamethasone is also used to treat 21-hydroxylase deficiency, especially when ACTH levels in the morning are relatively high. In this patient, his ACTH level, especially in the morning, was relatively high in relation to his cortisol level. Therefore, we started treatment with dexamethasone for this patient.

## 4. Conclusions

We experienced a case of congenital adrenal hyperplasia due to 21-hydroxylase deficiency which was diagnosed in a 31-year-old male who presented with infertility. Therefore, the possibility of 21-hydroxylase deficiency should be borne in mind, especially in subjects with infertility who were born before 1989 when newborn screening for 21-hydroxylase deficiency began.

## Figures and Tables

**Figure 1 diagnostics-13-00505-f001:**
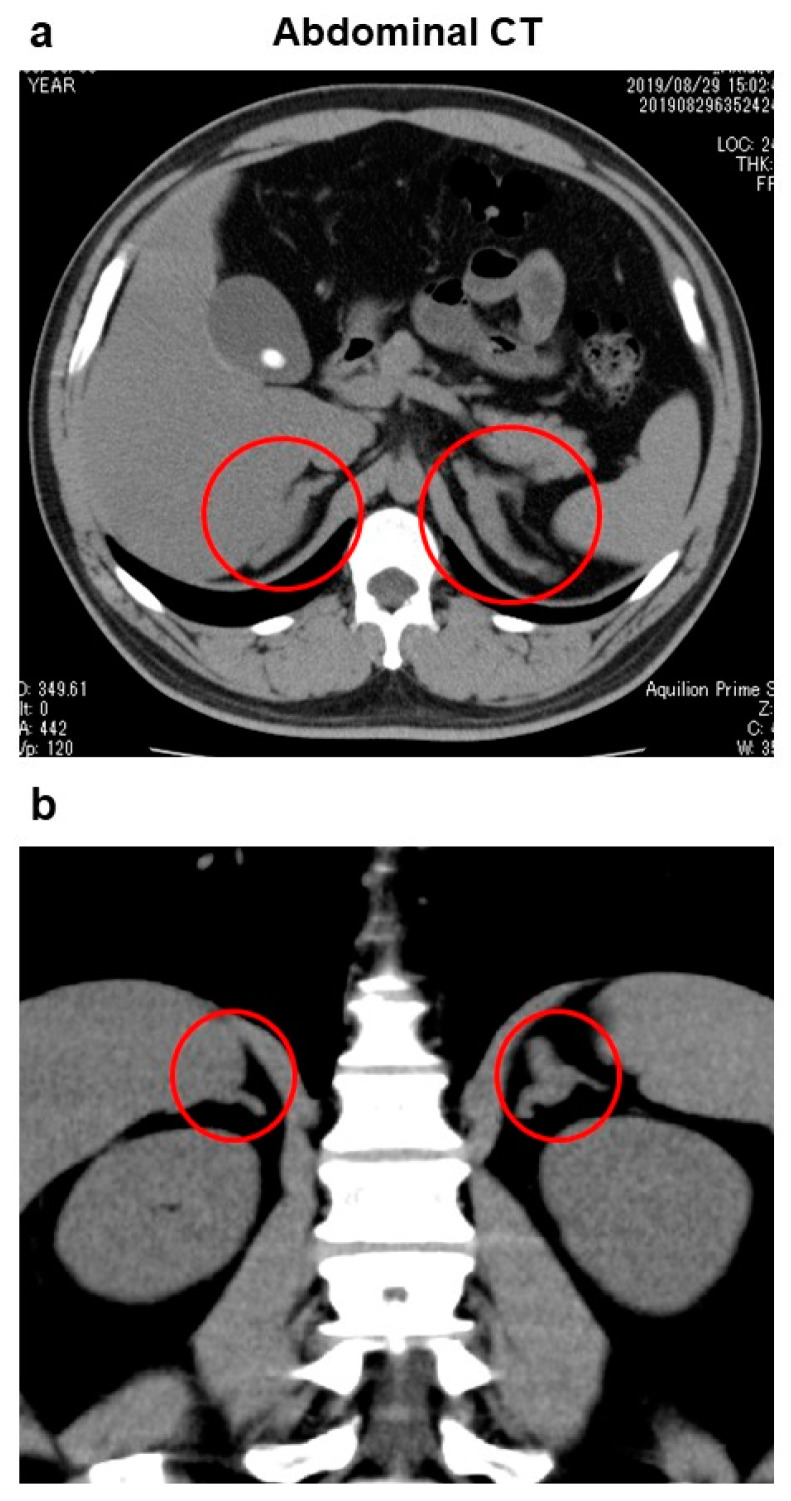
Abdominal computed tomography: (**a**) cross-sectional image, and (**b**) longitudinal image. Swelling was observed in bilateral adrenal glands, although morphology was normal (circles), suggesting the presence of adrenal hyperplasia. In addition, fatty liver, gallbladder stones and a right renal cyst were observed. There were no enlarged lymph nodes and no findings suggesting malignancy.

**Table 1 diagnostics-13-00505-t001:** Clinical parameters on admission in this 31-year-old subject.

Peripheral Blood		Blood Biochemistry		Endocrine Markers	
Red blood cells	571 × 10^4^/μL	Total protein	7.4 g/dL	TSH	3.62 μIU/mL
Hemoglobin	17.0 g/dL	Albumin	4.3 g/dL	FT3	3.47 pg/mL
Hematocrit	49.1%	Total bilirubin	0.5 mg/dL	FT4	1.18 ng/mL
White blood cells	6210/μL	AST	54 U/L	ACTH	172.2 pg/mL
Neutrophils	53.9%	ALT	123 U/L	Cortisol	6.9 μg/dL
Lymphocytes	35.3%	γ-GTP	66 U/L	DHEA-S	738 mg/dL
Monocytes	7.4%	LDH	189 U/L	GH	0.12 ng/mL
Eosinophils	2.4%	ALP	245 U/L	IGF-1	194 ng/mL
Basophils	1.0%	Creatinine	0.81 mg/dL	Prolactin	13.5 ng/mL
Platelets	21.6 × 10^4^/μL	BUN	20 mg/dL	LH	<0.10 mIU/L
Electrolytes		UA	6.2 mg/dL	FSH	0.22 mIU/mL
Sodium	137 mmol/L	CRP	0.15 mg/dL	Free testosterone	29.7 pg/mL
Potassium	4.0 mmol/L	Plasma glucose	109 mg/dL	Estradiol	63.1 pg/mL
Chloride	100 mmol/L	HbA1c	8.1%	Progesterone	8.15 ng/mL
Calcium	9.7 mg/dL	LDL-C	87 mg/dL	PRA	16.5 ng/mL/hr
Phosphorus	4.8 mg/dL	HDL-C	28 mg/dL	Aldosterone	659 pg/mL
Magnesium	2.2 mg/dL	Triglyceride	448 mg/dL	Adrenaline	16 pg/mL
				Noradrenaline	194 pg/mL

Abbreviation: AST, aspartate aminotransferase; ALT, alanine aminotransferase; γ-GTP, γ-glutamyl transpeptidase; LDH, lactate dehydrogenase; ALP, alkaline phosphatase; BUN, blood urea nitrogen; UA, uric acid; CRP, C-reactive protein; LDL-C, low density lipoprotein-cholesterol; HDL-C, high density lipoprotein-cholesterol; TSH, thyroid-stimulating hormone; FT3, free triiodothyronine; FT4, free thyroxine; ACTH, adrenocorticotropic hormone; DHEA-S, dehydroepiandrosterone sulfate; GH, growth hormone; IGF-1, Insulin-like growth factor; LH, luteinizing hormone; FSH, follicle-stimulating hormone; PRA, plasma renin activity.

**Table 2 diagnostics-13-00505-t002:** CRH, LHRH, GRH and TRH load test.

	Pre	15 Min	30 Min	60 Min	120 Min	180 Min
ACTH (pg/mL)	69.2	271.6	422.2	202.0	139.0	210.3
Cortisol (μg/dL)	8.8	8.9	8.7	8.1	8.9	8.9
LH (mIU/mL)	<0.1	0.43	0.55	0.46	0.42	
FSH (mIU/mL)	<0.1	0.31	0.38	0.57	0.79	
GH (ng/mL)	0.95	1.84	1.78	3.17	1.51	1.17
TSH (mIU/mL)	3.71	52.00	55.10	25.69	9.91	4.29
Prolactin (ng/mL)	8.2	88.3	65.7	25.3	19.0	

Abbreviation: CRH, corticotropin-releasing hormone; luteinizing hormone-releasing hormone; GRH, growth hormone-releasing hormone; TRH, thyrotropin-releasing hormone; ACTH, adrenocorticotropic hormone; LH, luteinizing hormone; FSH, follicle-stimulating hormone; GH, growth hormone; TSH, thyroid-stimulating hormone.

**Table 3 diagnostics-13-00505-t003:** Clinical parameters for steroid metabolism in this subject.

Urine Collection Test			Urine Steroid Fraction Analysis		
Parameter	Value	Reference Range	Parameter	Value	Reference Range
Cortisol	33.9 μg/day		Pregnanetriol	10.1 mg/gCre	0.0–0.3
Aldosterone	43 μg/day		11OH-androstenedione	22.5 mg/gCre	0.00–0.35
Pregnanediol	1.78 mg/day	0.16–0.79			
Pregnanetriol	36.12 mg/day	0.13–1.60	Other analysis		
17-KGS	0.67 mg/day	6.00–18.40	Parameter	Value	Reference range
(17-KS fraction)			17aOH-progesterone	52.4 ng/mL	0.0–0.6
Androsterone	7.18 mg/day	1.10–4.20	(Direct method)		
Etiocholanolone	1.82 mg/day	0.55–2.60	17aOH-progesterone	42.7 ng/mL	0.00–0.35
DHEA	4.42 mg/day	0.12–5.20	(Extraction method)		
11-ketoandrosterone	1.27 mg/day	0.00–0.12			
11-ketoetiocholanolone	3.00 mg/day	0.04–0.65			
11OH-androsterone	28.51 mg/day	0.40–2.30			
11OH-etiocholanolone	0.06 mg/day	0.03–0.65			

Abbreviation: 17-KGS, 17-ketogenic steroid; 17-KS, 17-ketosteroid; DHEA, dehydroepiandrosterone.

## Data Availability

Not applicable.

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
