# Peer review of "A Male Subject with Congenital Adrenal Hyperplasia due to 21-Hydroxylase Deficiency Which Was Diagnosed at 31 Years Old due to Infertility"

_diagnostics, 2023, doi:10.3390/diagnostics13030505_

Round 1

Reviewer 1 Report

The article is good.

The case report is well written

The literature is adequately described

Discussions are comprehensive

I recommend acceptance

Author Response

Thank you very much for your very favorable comments.

Reviewer 2 Report

Dear Authors,

The article is interesting. The case report is challenging. I have a few observations, please:

1.    Please unify the style and  the format of the paper starting with the abstract

2.    Abstract & Introduction - please use  “some enzymes” instead of “some enzyme”

3.    Abstract & Case presentation -  please use “adrenal insufficiency” instead of “hypoadrenalism”

4.    Abstract – The height should be introduced in terms of standard deviation from normal for age, sex, population

5.    Abstract – You should first introduce the clinical elements in terms of those positive clinical elements, and, eventually, of negative other aspects

6.    Abstract & Case presentation – Please provide the value of each hormone with normal ranges next to it since there are different methods of assessments

7.    Key words - You should introduce “male infertility”, as well

8.    Aim – Please use “We aim to introduce a male patient..:” instead of “Here we show..”

9.    Clinical presentation should be followed by the endocrine and imaging assessments

10. Legend Table 1 – Please provide the age of the patient at that moment since “admission” is anytime between the age of 31 and 37 years

11. Table 2 should be named such. The same goes for Table 3. That is why Table 2 is actually Table 4 and so on

12. ACTH stimulation test needs to be specified (the dose, the timeline of assays, the measurements, kits, etc.)

13. The Table with diurnal variations should be accompanied by some explanations and/or normal profile since many hormones, for instance, prolactin do not have a clinically relevance concerning their diurnal variations

14. Discussions is a section distinct from Conclusion

Thank you

Author Response

Response to Reviewer 2’s comments

The article is interesting. The case report is challenging. I have a few observations, please:

Please unify the style and the format of the paper starting with the abstract

We unified the style and format of the paper starting the abstract.

Abstract & Introduction - please use “some enzymes” instead of “some enzyme”

We used the expression “some enzymes” in the revised version.

Abstract & Case presentation - please use “adrenal insufficiency” instead of “hypoadrenalism”

We used the expression “adrenal insufficiency” in the revised version.

Abstract – The height should be introduced in terms of standard deviation from normal for age, sex, population

Thank you very much for valuable suggestion. According to your kind suggestion, we added this point in the revised version.

Abstract – You should first introduce the clinical elements in terms of those positive clinical elements, and, eventually, of negative other aspects

Thank you very much for valuable suggestion. According to your kind suggestion, we amended this point in the revised version.

Abstract & Case presentation – Please provide the value of each hormone with normal ranges next to it since there are different methods of assessments

Thank you very much for valuable suggestion. According to your kind suggestion, we amended this point throughout the manuscript in the revised version.

Key words - You should introduce “male infertility”, as well

We included the expression “male infertility” in key words.

Aim – Please use “We aim to introduce a male patient..:” instead of “Here we show..”

According to your kind suggestion, we amended this point in the revised version.

Clinical presentation should be followed by the endocrine and imaging assessments

According to your kind suggestion, we amended this point in the revised version.

Legend Table 1 – Please provide the age of the patient at that moment since “admission” is anytime between the age of 31 and 37 years

The description “37-year-old” was a mistake. We amended this mistake in the revised version. This patient was hospitalized when he was 31 years old. In addition, we added the age in Table 1.

Table 2 should be named such. The same goes for Table 3. That is why Table 2 is actually Table 4 and so on

We amended this point in the revised version.

ACTH stimulation test needs to be specified (the dose, the timeline of assays, the measurements, kits, etc.)

Thank you very much for your valuable suggestion. According to your kind suggestion, we added this point in the revised version.

The Table with diurnal variations should be accompanied by some explanations and/or normal profile since many hormones, for instance, prolactin do not have a clinically relevance concerning their diurnal variations

Thank you very much for your valuable suggestion. According to your kind suggestion, we added the explanation about this point in the revised version.

Discussions is a section distinct from Conclusion

We separated conclusion section from discussion section in the revised version.

Reviewer 3 Report

The case report submitted by Kaneto H et al. addressed “ A male subject with congenital adrenal hyperplasia due to 21-hydroxylase deficiency which was diagnosed at 31 years old due to infertility.” The report's relevance was highlighted well, and no breaches in ethical practice were noted. This case report is well written and structured and adds to current knowledge about congenital adrenal hyperplasia, but reports need major formatting, particularly in the Table section.

The authors should mention the family history of the patient. Since the disease authors are discussing is one of the autosomal recessive disorders, kindly note if there was any history of the disease in the family.

The authors also explain any patient outcome and follow-up after starting medication.

Minor comments:

Figure 1: Please explain figures 1a and b in detail. The arrows are not unambiguous.

Author Response

Response to Reviewer 3’s comments

The case report submitted by Kaneto H et al. addressed “ A male subject with congenital adrenal hyperplasia due to 21-hydroxylase deficiency which was diagnosed at 31 years old due to infertility.” The report's relevance was highlighted well, and no breaches in ethical practice were noted. This case report is well written and structured and adds to current knowledge about congenital adrenal hyperplasia, but reports need major formatting, particularly in the Table section.

The authors should mention the family history of the patient. Since the disease authors are discussing is one of the autosomal recessive disorders, kindly note if there was any history of the disease in the family.

Thank you very much for your valuable suggestion. According to your kind suggestion, we added the description about this point in the revised version of the manuscript (page 7, lines 1-5)

The authors also explain any patient outcome and follow-up after starting medication.

Thank you very much for your valuable suggestion. According to your kind suggestion, we added the description about this point in the revised version of the manuscript (page 10, lines 4-8)

Minor comments:

Figure 1: Please explain figures 1a and b in detail. The arrows are not unambiguous.

Thank you very much for your valuable suggestion. According to your kind suggestion, we added the description about this point in the revised version of the manuscript (page 8, lines 1-4 from the bottom, page 9, line 1, page 18, lines 1-5). In addition, we put red circles instead of arrows in Figure 1 so that readers can easily understand the content.